# Leaching of Polycyclic Aromatic Hydrocarbons from the Coal Tar in Sewage Wastewater, Acidic and Alkaline Mine Drainage

**DOI:** 10.3390/ijerph19084791

**Published:** 2022-04-15

**Authors:** Jean Bedel Batchamen Mougnol, Frans Waanders, Elvis Fosso-Kankeu, Ali Rashed Al Alili

**Affiliations:** 1Water Pollution Monitoring and Remediation Initiatives Research Group, Centre of Excellence in Carbon-Based Fuels, North-West University, Potchefstroom 2520, South Africa; frans.waanders@nwu.ac.za; 2Department of Mining Engineering, Florida Science Campus, College of Science Engineering and Technology, University of South Africa, Pretoria 0002, South Africa; elvisfosso.ef@gmail.com; 3DEWA R&D Center, Dubai Electricity and Water Authority (DEWA), Dubai P.O. Box 564, United Arab Emirates; alialalili@gmail.com

**Keywords:** PAH pollutants, acid mine drainage, alkaline mine drainage, sewage wastewater

## Abstract

Polycyclic aromatic hydrocarbons (PAHs) have been a problem in the environment for an extended period. They are mostly derived from petroleum, coal tar and oil spills that travel and are immobilized in wastewater/water sources. Their presence in the environment causes a hazard to humans due to their toxicity and carcinogenic properties. In the study, coal tar was analyzed using Gas Chromatography–Mass Spectrometry (GC–MS) and a concentration of 787.97 mg/L of naphthalene, followed by 632.15 mg/L of phenanthrene were found to be in the highest concentrations in the various water sources such as sewage, alkaline and acid mine drainage. A design column was used to investigate the leaching process and assessments were conducted on 300 mL of the various water sources mentioned, with 5 g of coal tar added and with monitoring for 4 weeks. The influence of the physiochemical properties of the receiving water sources, such as sewage, and acid and alkaline mine drainage, on the release of PAHs from the coal tar was assessed. The acidic media was proven to have the highest release of PAHs, with a total concentration of 7.1 mg/L of released PAHs, followed by 1.2 mg/L for the sewage, and lastly, 0.32 mg/L for the alkaline mine drainage at room temperature.

## 1. Introduction

The world’s largest energy consumption is produced by coal, making it the second-largest energy source [1]. South Africa (SA) has been ranked the fifth producer and consumer of coal and 86% of the energy produced in SA is derived from it [1]. In SA, coal has been one of the reasons behind the country’s improved financial and economic growth due to its availability and cost-effectiveness. The rise in energy demand has exposed the use of coal as a sustainable energy source which causes a potential health hazard to humans and the environment due to the release of a by-product known as coal tar [2]. Thousands of coking processes are responsible for the release of coal tar and gas fuel production, which are being derived from coal transformation [3].

Figure 1 represents the usage of coal in SA from the year 1980 to 2015 and it has the potential to release a by-product that consists of polycyclic aromatic hydrocarbons (PAHs).

Coal tar is a by-product of coal that is released from various processes of the coal gas and coking process. It is semi-liquid in nature with a density of 1.15 to 1.4 g/cm^3^ and has a foul smell [4].

Polycyclic aromatic hydrocarbons (PAHs) are a group of organic compounds that are life-threatening because of their chemical structures, that consist of benzene rings bonded in a linear and angular position [5]. Though there are hundreds of PAH compounds, only 16 PAHs are commonly identified in various wastewater and water sources [6]. See also Table 1.

PAHs are released from various industries, from petroleum, coal tar and oil spills that end up settling in the environment and water streams, making them hazardous due to their toxicity level [7]. According to Wlodarczyk-Makula [8], the leaching of the pollutants in the various water sources contain PAHs with different chemical properties and concentration ranges from 46 to 70 g/d. Świetlik et al. [9] investigated the photodegradation of PAHs in distilled water and river water. Before the experiment, solids were removed from the river water using a nylon membrane with an opening size of 0.45 µm to prevent any reaction that may occur between the PAHs and the water. It was found that the PAHs were more concentrated in river water compared to distilled water. This was attributed to the physiochemistry of the river water, the temperature and the pH level [9].

Based on the major effluents that occur in various industries, PAHs have attracted a lot of attention due to their presence in wastewater and other water sources. Their deposition into the environment has led to major health concerns such as carcinogenic, mutagenic and teratogenic symptoms. The solubility of PAHs is dependent on various factors such as the temperature, pH, ionic strength and water matrix of the dissolved organic carbon [10]. It has been reported by Mojiri, Zhou, Ohashi, Ozaki and Kindaichi [5] that the concentration of pyrene in water in South Africa is about 1,118,000 × 10^−6^ mg/L and 8,310,000 × 10^−6^ mg/L for benzo(b)fluoranthene in sewage wastewater.

This study aims to assess the influence of the physiochemical properties of receiving water sources, such as sewage, and acid and alkaline mine drainage, on the release of PAHs from coal tar. Table 2 shows the molecular weight of the PAHs that are mostly found in water sources and recognized by the World Health Organization (WHO). A few PAHs have been identified to be life-threatening due to their extreme effects on humans, as established by their chemical properties depicted in Table 2.

It has been reported by Mojiri, Zhou, Ohashi, Ozaki and Kindaichi [5] that, in an aquatic environment, the PAHs concentrations were 0.03 × 10^−6^ mg/L to 83,100,000 × 10^−6^ mg/L. Among these PAHs, the lower molecular weight species, such as naphthalene, which consist of two to three rings, are not as cancerous as other PAHs of four to ten ring structures, such as chrysene, pyrene and others [11,12].

Table 3 shows the PAHs content in various water sources. However, drinking water was pointed out to have a higher PAHs presence compared to groundwater.

During the leaching process, PAHs normally migrate into the soil surface and groundwater where most of the PAHs are adsorbed or biodegraded in the existing minerals or enzymes in the soil. The remaining PAHs then largely channel into the drinking water resulting in it seeming to be more polluted than the groundwater [13,14].

### Phase Partition Equilibrium (Raoult’s Law) and Dissolution of Kinetics

Raoult’s law can be applied for the water samples and the non-miscible organic phase (PAHs), both having an ideal behaviour [15] that best fits the study aim. The ideal behaviour happens to be linear, whereby the more the coal is immersed in the water deposit, will result in more PAHs released in that water. Raoult’s law [16] can best describe the influence on the leaching process of various parameters; however, only pH and contact time were investigated in the study to determine their influence on the leaching process.

The distribution of the pollutant “coal tar”, that contains PAHs, into water sources (sewage wastewater, acidic and alkaline mine drainage) is controlled by the molecular diffusion of the coal tar interphase [17]. Since the viscosity of the coal tar is also considered, the global mass transfer is controlled by the slowest rate [18]. This can be attributed to the release of the PAHs in the water source by ageing. The slower the diffusion of the (coal tar) water interface, the higher the concentration of PAHs in the water source [3,14].

## 2. Materials and Methods

In the present study, the acid mine drainage was collected at Witbank, South Africa, from a river near a large mine; the alkaline mine drainage was collected at a water treatment plant at Middleburg, South Africa; and the sewage wastewater was collected in Cape Town Municipality, South Africa. The coal tar was obtained at a coking oven plant in New Castle KwaZulu-Natal, South Africa.

### 2.1. Research Methodology

During the leaching process, 5 g of coal tar/water sources were inserted in the columns, having dimensions of 300 cm by 8 cm internal diameter (Figure 1). The columns were filled with 300 mL of the lixiviant of each water source for a 4-week period. Samples were collected weekly using a 50 mL syringe with a syringe filter size of 0.45 µm, to prepare the PAHs for analysis. The system was kept away from light, and it was covered with aluminium foil to prevent light from penetrating. The results of the aliquot were triplicated and analyzed using Gas Chromatography–Mass Spectrometry (GC–MS) with a Thermo Scientific (TSQ 8000) (Cape Town, South Africa) Triple Quadrupole MS.

Figure 2 shows the leaching columns for the three water sources. The acid mine drainage column was observed to have a strong visual colour change (yellowish) compared to the alkaline and sewage water source. The colour shows that a strong interaction between the coal tar and the acid mine drainage occurred, which strongly reveals the release of the PAHs.

#### Determination of PAHs from GC–MS

There are various techniques such as HPLC-UV, GC-FID and GC–MS–MS or GC–MS that can be used to analyze PAHs. Since GC–MS has been regarded internationally as the gold standard analytical technique, with the ability to analyze and detect tiny amounts of particles in a substance, as reported by Wise, et al. [19], it was used in the study.

An SPE cartridge (SDBL-100 µm Styrene-divinylbenzene) was used to determine the PAHs content in the coal tar and their presence in a leachate. Half volumes of 12 mL of methanol and 10 mL of deionised water were used, and the remaining halves were added with the sample (coal tar) for calibration. Pressure was applied for about 15 min until the solution with the sample was dried completely.

A standard solution was prepared:

Sodium sulphate was added to a 10 mL Pyrex tube and 2 mL of ethyl acetate was eluted and pressure was applied until the ethyl acetate was transferred into the Pyrex tube. The tube was closed, then vortexed, and 150 µL was measured and then transferred into the GC vial, which was crimped and vortexed.

The prepared standard solution was then used for PAHs analysis using GC–MS.

## 3. Results and Discussion

In Table 4, the results are shown for the water sources analyzed in the study.

Figure 2 represents the PAHs content in the coal tar before the leaching process was conducted. It was found that, from the coal mass of 116.3 mg with 2 mL of hexane added as a solvent, when the PAHs were observed using the GC–MS technique, a total of 16 PAHs were detected. Naphthalene, followed by phenanthrene, fluoranthene and acenaphthylene, were found to be in the highest concentrations, compared to the remaining PAHs. The overall PAHs concentration, identified from the coal tar used, was found to be 2916.47 mg/L.

Naphthalene and phenanthrene were the simplest and most water-soluble molecules of the PAH group observed, and this serves as a model for the dissolution of the light hydroxylated aromatic compounds. The highest dissolution occurred in the acid mine drainage sample. The transfer kinetics of pollutants from the coal tar in the water was the highest at the lower molecular weights of PAHs. The experimental results of this study also showed ageing between the coal tar and the various water samples, resulting in the progressive release of PAHs. The deposit of coal tar in the acidic mine drainage should be given more attention, as PAHs were proven to be released most effectively with the acid mine drainage, which prompted the release of PAHs hazardous to the environment and to humans. According to Sharma and Lee [20], naphthalene was found to be the most abundant PAH in the coal tar, which was also proven in the present study, as can be seen in Figure 3.

According to Makelane et al. [21], pyrene was found to be most concentrated in sewage wastewater, which was also proven in the present study.

All the results on the leaching process of the PAHs from the various water sources are tabulated in Table 5, Table 6 and Table 7. A total of 16 PAHs were detected in the three water sources; however, acenaphthene was not detected in any of the three water sources used in the present study, albeit a concentration of 18.46 mg/L was found when the coal tar was analyzed. It was noted that the concentration of the PAHs increased with the ageing process. Ageing (or time) was noted to be a factor to be considered in the physiochemical conditions between the water source and the PAHs. The presence of the PAHs in the water source increases as ageing increases. The lower molecular weight PAHs, with two to three rings, were identified to be highly soluble in the acidic media and slightly soluble in the sewage wastewater, and less soluble in the alkaline media (Table 6). Since PAHs are non-polar and have no charge, the electron pairs on the aromatic rings play a role in the physiochemical reaction of the PAHs and the water forming radicals’ reaction that triggered the PAHs to be released. Table 5, Table 6 and Table 7 represent the experimental results obtained during the leaching process. It should also be noted that all experiments were conducted in triplicate.

Figure 4 represents the PAHs dissolution in various water sources and how effective a role Raoult’s law plays in the leaching process. The immersion of coal tar in the various water sources used, with their different physiochemistry, resulted in obtaining different PAHs values, as shown in Figure 4. According to Boulangé, Lorgeoux, Biache, Michel, Michels and Faure [3], there are factors such as temperature, ionic strength, pH and dissolved organic carbon. However, in this study, the pH and the dissolved organic carbon were considered in the leaching. There is a need to understand the driving forces of these factors:

The dissolved organic carbon concentration can be influenced by pH [22]; the effect of pH does not influence the neutral organic compound. Neutral organic compounds contain functional groups such as hydroxyl, ether, ketone, lactone, aldehyde and ester because of the oxygen present in them, which can easily react with water to form hydrogen bonds. The presence of these functional groups increases the solubility in water [3].

This study does agree with Boulangé, Lorgeoux, Biache, Michel, Michels and Faure [3], as it labelled the nature of the dissolved PAHs as being acidic by having them dissolved or favored in an acidic mine drainage at a lower pH.

### Influence of Ageing of Coal Tar/Various Water Sources Interphase

Figure 5 represents how ageing influenced the release of the different PAHs and the different water sources. The ageing phenomenon of the coal tar/water sources interphase was found to be an important aspect that has a significant consequence for the long-term fate of coal tar in various water sources, with an increase in the release of higher PAHs concentrations. During the leaching process, anthracene was highly dissolved in the acidic mine drainage and sewage wastewater at week 4. The ageing of PAHs in sewage wastewater agreed with Cai, Ding, Zhang, Wang, Wang, Ren and Dong [14], who mentioned that the longer PAHs are in water, the more they reach an unacceptable level over time. Since time is directly proportional to ageing, as time increases, the PAHs concentration in the coal tar availability reduces, resulting in a greater dissolution.

Since coal is still currently being used as one of the most substantial sources of energy in SA, there is still a prediction of a large release of coal tar that also exhibits PAHs. An intensive consideration should be implemented to avoid coal tar being channeled into various waters. The processes involving the transformation of coal should be conducted far away from water sources, allowing ageing to occur between the by-product (coal tar) and the soil. In doing so, the exposure of PAHs on the surface will be no risk to humans, as it was reported by Patel et al. [23] that the concentrations of 1 ng/L and 11 μg/L of PAHs in drinking water, as per the WHO regulations, are acceptable.

Since billions of rands have and still are being used on coal for energy consumption and production, millions of rands also play a part in remediating the PAHs from the environment and water.

## 4. Conclusions

In coal tar-contamination in various water sources, the main mechanism involved in the PAHs release is known to be dissolution; this is described by Raoult’s law, which states that the equilibrium concentration of a compound in a water source is a function of the compound’s water solubility and its molar fraction in the initial phase in contact with water. The acidic media was proven to have the highest release of PAHs, with a total concentration of 7.1 mg/L, followed by 1.2 mg/L for the sewage, and lastly, 0.32 mg/L for the alkaline mine drainage. Furthermore, it was found that the lower molecular weight PAHs were more soluble in the acidic water as it resulted in releasing a higher PAH concentration.

During the leaching on the acidic mine drainage, it was observed that a pH of 2.77 and dissolved organic carbon of 2.25 mg/L, resulted in the release of 7.1 mg/L of PAHs from the coal tar. The reaction of the π–π bonding also plays a vital role in the reaction of the aromatic compound of the PAHs with the acidic water molecules. PAHs happened to be more soluble in the acidic mine drainage compared to the alkaline, and slightly soluble in sewage depending on the pH and the dissolved organic carbon. Since PAHs are non-polar and only expected to be soluble in an aromatic’s solvent however, the pH and dissolved organic carbon are factors to be considered when anticipating the solubility of PAHs in wastewater and water sources.

## Figures and Tables

**Figure 1 ijerph-19-04791-f001:**
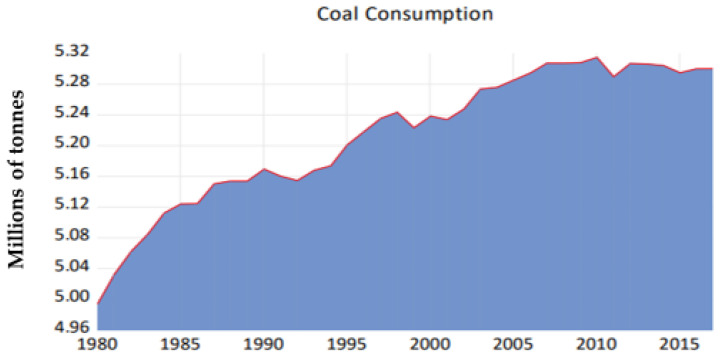
Coal Consumption in SA from 1980 to 2015.

**Figure 2 ijerph-19-04791-f002:**
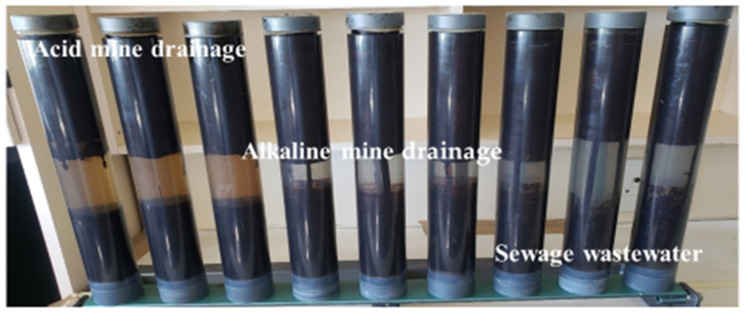
Leaching process of PAHs in acid mine drainage, alkaline mine drainage and sewage wastewater.

**Figure 3 ijerph-19-04791-f003:**
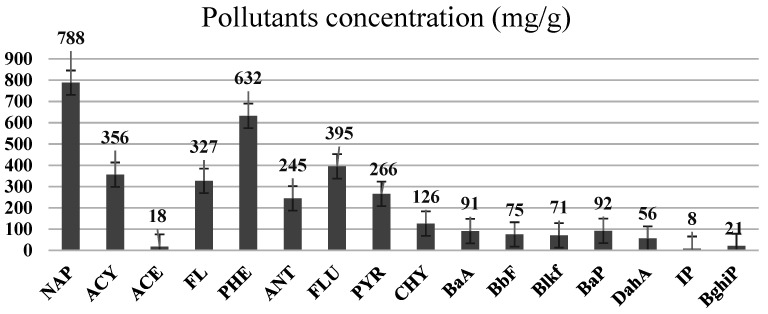
The concentration of PAHs in the coal tar sample, as found in the present study.

**Figure 4 ijerph-19-04791-f004:**
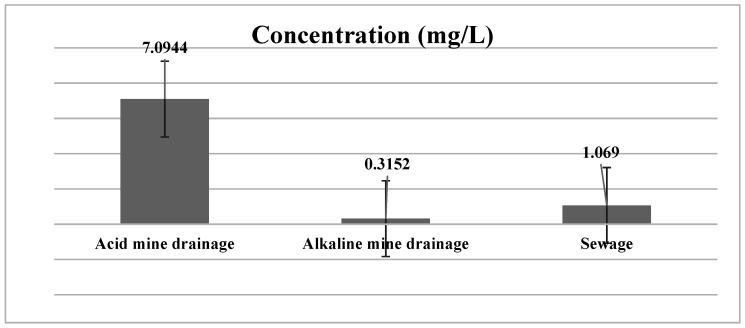
The concentration of total PAHs in the three different water samples studied.

**Figure 5 ijerph-19-04791-f005:**
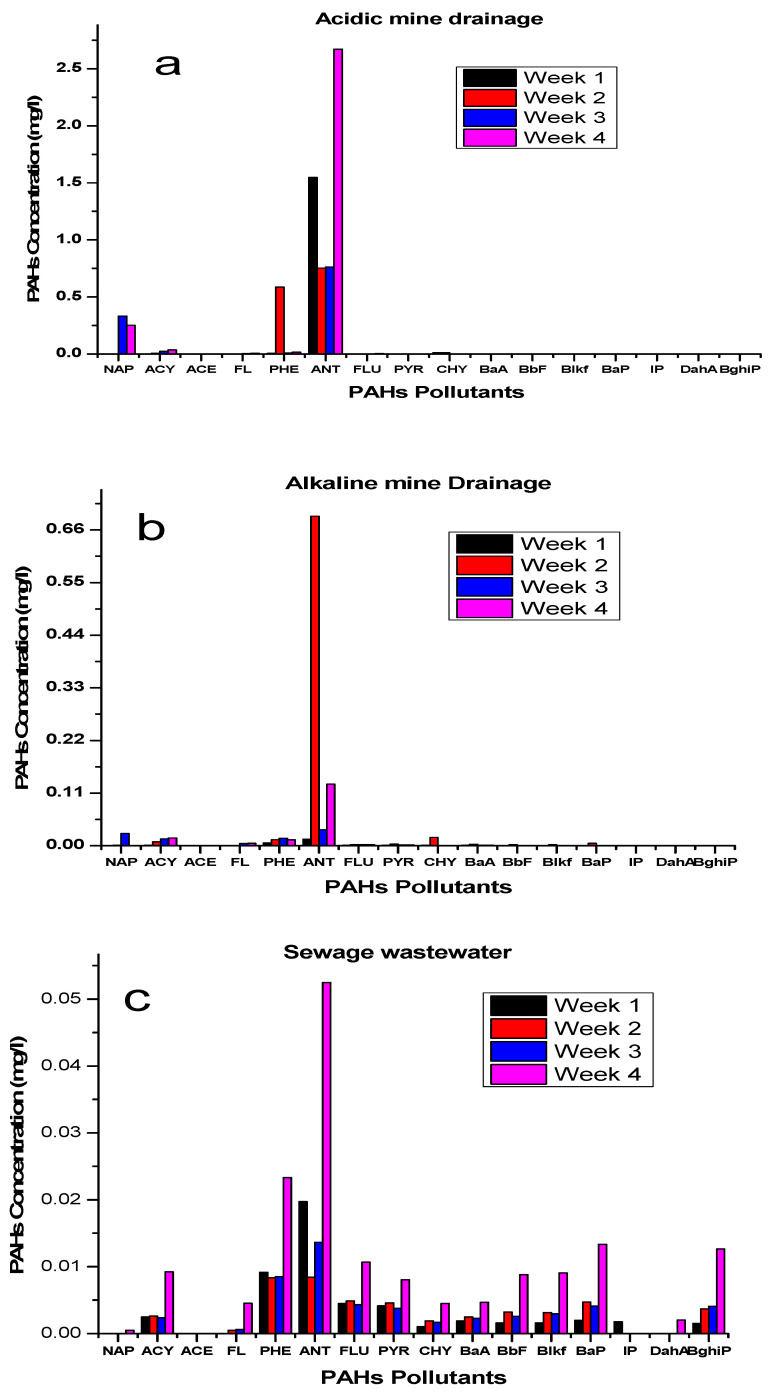
Ageing of PAHs in acidic medium (**a**), alkaline medium (**b**), and sewage medium (**c**).

**Table 1 ijerph-19-04791-t001:** Acronyms of the PAHs investigated in this study.

Polycyclic Aromatic Hydrocarbons (PAHs)	Acronyms
Naphthalene	NAP
Acenaphthylene	ACY
Acenaphthene	ACE
Fluorene	FL
Phenanthrene	PHE
Anthracene	ANT
Fluoranthene	FLU
Pyrene	PYR
Chrysene	CHY
Benzo(a)anthracene	BaA
Benzo(b)fluoranthene	BbF
Benzo(k)fluoranthene	Blkf
Benzo(a)pyrene	BaP
Dibenzo(a,h)anthracene	DahA
Indeno(1,2,3,c-,d-)pyrene	IP
Benzo(g,h,i)pyrene	BghiP

**Table 2 ijerph-19-04791-t002:** Chemical and structural information of a few common PAHs compounds. Reprinted/adapted with permission from [11]. Copyright 2020, copyright owner’s Van-Huy Nguyen.

PAHs	Formula	No. of Rings	Molar Weight (g/mol)	Geometry
NAP	C_10_H_8_	2	128	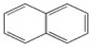
ANT	C_14_H_10_	3	178	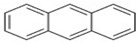
PHE	C_14_H_10_	3	178	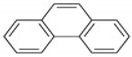
CHY	C_18_H_12_	4	228	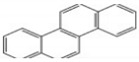
CHY	C_16_H_10_	4	202	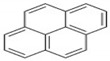
BaP	C_20_H_12_	5	252	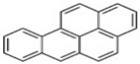
BghiP	C_22_H_12_	6	276	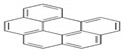
ACY	C_12_H_8_	3	152	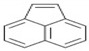
FL	C_13_H_10_	3	166	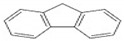
FLU	C_16_H_10_	4	202	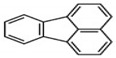
BaA	C_18_H_12_	4	228	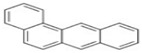
BbF	C_20_H_12_	5	252	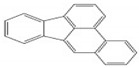
Blkf	C_20_H_12_	5	252	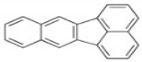
DahA	C_22_H_14_	5	278	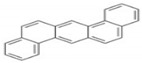
BaP	C_22_H_12_	6	276.33	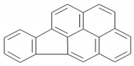
ACE	C_12_H_10_	5	154.21	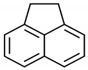

**Table 3 ijerph-19-04791-t003:** The PAHs content in various water sources. Reprinted/adapted with permission from [11]. Copyright 2020, copyright owner’s Van-Huy Nguyen.

PAHs (×10^−6^) mg/L	ACE	ACY		ANT	BaA	BaP	BbF	Blkf	BghiP	CHY	DahA	FL	FLU	IP	NAP	PHE	PYR
**Drinking Water**	3.8 to 478	1.8 to 1210		1.4 to 71	2.29 to 10	1.3 to 8	2.1 to 24	4.6 to 24	2.0 to 8	1.8 to 27	2.0 to 9	4.0 to 41,000	6.5 to 1,430,000	1.6 to 3	4.6 to 14,000	13.1 to 139,000	4.2 to 92,000
**Rivers and Lakes**	2.6 to 579,000	2.7 to 537,000		1.0 to 256,000	0.6 to 3200	0.5 to 1,239,000	1.2 to 7,800,000	0.8 to 3100	0 to 11,700	1.8 to 4300	4.0 to 11,400	5.6 to 2,480,000	4.2 to 2,498,000	1.0 to 7200.0	52.5 to 6900	13.3 to 126,000	2.9 to 1,138,000
**Groundwater**	0.4 to 149	0.8 to 13		0.1 to 196	0.1 to 6	3.0 to 123	1.9 to 39	5.1 to 30	0.4 to 9	0.1 to 71	0.1 to 4	0.4 to 168	2.0 to 51	3.6 to 12	2.1 to 281	2.0 to 179	0.3 to 42
**Wastewater**	28.8 to 100	16.6 to 66		42.0 to 295	46	71.6 to 1,447,000	82.0 to 8,310,000	100.0 to 204	92	20.7 to 112	0	20.0 to 234,000	14.0 to 2,340,000	21	40.0 to 47,000	33.0 to 6,495,000	19.1 to 1,186,600
**Seawater**	2.6 to 4200	4.5 to 4100	0.1 to 3350.0	0.0 to 17,490	0.0 to 17,490	0.2 to 28,490	0.0 to 32,050		0.2 to 14,790	0.1 to 42,710	0.0 to 32,340	0.2 to 1520	0.0 to 6610	0.0 to 46,600	75.9 to 7800	0.2 to 1080	0.0 to 987
**Sediments**	0.6 to 1821	1.7 to 13	2.0 to 658	0.2 to 152	0.0 to 739	<1 to 932	3.8 to 17,486		8.9 to 5153	0.9 to 193	1.8 to 999	0< to 52	<1 to 24,857	0.4 to 552	<1 to 69	5.7 to 410	2.8 to 27

**Table 4 ijerph-19-04791-t004:** Acid mine drainage, alkaline mine drainage and sewage wastewater analysis.

Water Samples	Physical/Aesthetic Parameters	Unit Result	Result
Acid mine drainage	Acidity as CaCO_3_	mg/L	672
	Alkalinity-Total as CaCO_3_	mg/L	<10.0
	Dissolved Oxygen	mg/L	8.00
	Electrical Conductivity @ 25 °C	mS/m	214
	pH @ 25 °C	pH units	2.77
	**Organic Parameters**		
	Dissolved organic carbon	mg/L	2.25
Alkaline mine drainage	Acidity as CaCO_3_	mg/L	12.0
	Alkalinity-Total as CaCO_3_	mg/L	402
	Dissolved Oxygen	mg/L	8.30
	Electrical Conductivity @ 25 °C	mS/m	352
	pH @ 25 °C	pH units	7.95
	**Organic Parameters**		
	Dissolved organic carbon	mg/L	4.35
Sewage wastewater	Acidity as CaCO_3_	mg/L	216
	Alkalinity-Total as CaCO_3_	mg/L	438
	Dissolved Oxygen	mg/L	0.50
	Electrical Conductivity @ 25 °C	mS/m	154
	pH @ 25 °C	pH units	6.70
	**Organic Parameters**		
	Dissolved organic carbon	mg/L	113

**Table 5 ijerph-19-04791-t005:** The recovery of PAHs in an acidic mine drainage @ 25 °C.

	PAHs Content	Week 1 (mg/L)	Week 2 (mg/L)	Week 3 (mg/L)	Week 4 (mg/L)	PAHs Concentration (mg/L)
Acid Mine Drainage	NAP	0	0	0.332402	0.254163	0.5877
	ACY	0.00232	0.006527	0.023753	0.03721	0.0707
	ACE	0	0	0	0	0
	FL	0	0	0.005651	0.009247	0.0155
	PHE	0.0074534	0.585888	0.0095059	0.0173442	0.620
	ANT	1.548783	0.753182	0.763863	2.669656	5.7355
	FLU	0.0015533	0.001267	0.00161	0.00473	0.0092
	PYR	0.001593	0.001553	0.001507	0.003643	0.0083
	CHY	0.013611	0.013096	0	0.001075	0.0288
	BaA	0.0012033	0.0011733	0.0012	0.0018733	0.0054
	BbF	0.000393	0.000347	0.00037	0.002063	0.0032
	Blkf	0.0001967	0.0001367	0.0001643	0.0019023	0.0024
	BaP	0.00331	0.002507	0.000253	0.002853	0.0098
	IP	0	0	0	0	0
	DahA	0	0	0	0	0
	BghiP	0	0	0	0.002315	0.0023
PAHs Total						7.0944

**Table 6 ijerph-19-04791-t006:** The recovery of PAHs in an alkaline mine drainage @ 25 °C.

	PAHs Content	Week 1 (mg/L)	Week 2 (mg/L)	Week 3 (mg/L)	Week 4 (mg/L)	PAHs Concentration (mg/L)
Alkaline mine drainage	NAP	0	0	0	0.000483	0.0005
	ACY	0.002493	0.00264	0.002393	0.009223	0.0167
	ACE	0	0	0	0	0
	FL	0	0.000506	0.000633	0.004562	0.0057
	PHE	0.0091672	0.0083211	0.00851	0.0233266	0.0493
	ANT	0.019723	0.008446	0.013635	0.052464	0.0943
	FLU	0.004533	0.004873	0.004317	0.010687	0.0244
	PYR	0.004137	0.00459	0.00377	0.008077	0.0206
	CHY	0.001077	0.00192	0.001702	0.004518	0.0092
	BaA	0.0019267	0.00252	0.0022933	0.0046833	0.0114
	BbF	0.001637	0.00322	0.002577	0.00884	0.0163
	Blkf	0.00163	0.003142	0.002957	0.0090683	0.0168
	BaP	0.001993	0.004723	0.004103	0.013353	0.0242
	IP	0.001769	0	0	0	0.0018
	DahA	0	0	0	0.002039	0.0020
	BghiP	0.001521	0.003682	0.00409	0.012656	0.0219
Total PAHs						0.3152

**Table 7 ijerph-19-04791-t007:** The recovery of PAHs in sewage wastewater @ 25 °C.

	PAHs Content	Week 1 (mg/L)	Week 2 (mg/L)	Week 3 (mg/L)	Week 4 (mg/L)	PAHs Concentration
						(mg/L)
Sewage Wastewater	NAP	0	0.00096	0.02564	0.02579	0.0524
	ACY	0.001167	0.008353	0.014387	0.016147	0.0401
	ACE	0	0	0	0	0
	FL	0	0	0.004661	0.005021	0.0097
	PHE	0.005983	0.012669	0.015708	0.012209	0.0466
	ANT	0.013944	0.688642	0.033595	0.128699	0.8649
	FLU	0.001	0.00263	0.002653	0.002507	0.0088
	PYR	0.001223	0.00338	0.001847	0.00194	0.0084
	CHY	0.001336	0.01732	0	0.000317	0.0190
	BaA	0.001	0.00318	0.001023	0.001303	0.0066
	BbF	0.000343	0.00235	0.00035	0.00064	0.0037
	Blkf	0.0001347	0.002133	0.0001453	0.0004623	0.0029
	BaP	0.00024	0.00511	0.000257	0.0006	0.0063
	IP	0	0	0	0	0
	DahA	0	0	0	0	0
	BghiP	0	0	0	0	0
Total PAHs						1.0690

## Data Availability

Not applicable.

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
