# Peer review of "Leaching of Polycyclic Aromatic Hydrocarbons from the Coal Tar in Sewage Wastewater, Acidic and Alkaline Mine Drainage"

_ijerph, 2022, doi:10.3390/ijerph19084791_

Round 1

Reviewer 1 Report

The work presented in the manuscript is interesting, but the presentation of results and the discussion should be improved.

General:

Please check and harmonize figures layout (font, dimension of characters, etc). Also, be more precise in captures

Intro:

please explain why did you select  coal tar over other sources

p3- line 63: Please explain more clearly why drinking water contains more PAH than groundwater

Methods:

Do the temperature has a role? was it controlled somehow during the experiments?

Indicate in Methods you also analysed the coal tar by gas chromatography

Results and discussion:

this section must be improved. please present the results in a more structured and logic way, discussing the influence of each of the investigated parameters.

p7, Lines 145 - 149: that is not clear, plaese explain better giving some examples

P8 line 159: the sentence is not clear

Conclusions:

The conclusions report results and considerations that were not presented or discussed in the results section, please check and rewrite the result section more completely and logically

p. 9 line 174: the sentence is not clear

Reviewer 2 Report

  1. Please use SI-compliant units.
  2. The authors should re-edit the text to make it easier to read, for example, the tables should not slip apart and the concentrations should be in one row in Table 3.
  3. in line 172, "a" was omitted from "that"

Reviewer 3 Report

Review

Comments to the Authors:

This manuscript presents studies on the effects of leaching of polycyclic aromatic hydrocarbons from the coal tar in sewage wastewater, acidic and alkaline mine drainage. The research achievements are interesting and promising. On the other hand, after a careful and detailed analysis there are some suggestions and questions to the authors. Thus, it is recommended to read all manuscript carefully and make corrections.

  • Page 1, line 18: It is recommended to put a space between the number and the unit in the whole manuscript: 5g.
  • Pages 2 - 3, lines 57 – 58: The parameters of aromatics shown in Table 2 are widely available in the literature.Hence, is it necessary to present this data in this manuscript?In my opinion, table 2 can be deleted.
  • Page 4, lines 79 – 80: In my opinion, a reference should be added at the end of the paragraph. The reference should be related to the statement: ‘The slower the diffusion of (coal tar) water interface, the higher the concentration of PAHs in the water source.’
  • Page 4, lines 81 – 101: It is recommended to divide the section ‘Materials and Methods’ into other subsections, e.g. ‘Characterization of the research materials’ and ‘Research methodology’.
  • Page 4, lines 81 – 101: It is recommended to explain the research methods in detail. More details should be provided on the Gas Chromatography-Mass Spectrometry (GC-MS)
  • Pages 4 – 8, lines 102 – 164: In my opinion, in the 'Results and discussion' section, a broader discussion of the obtained research results and their comparison with similar results available in the literature should be performed.
  • Pages 4 – 8, lines 102 – 164: It seems to me that the authors should emphasize the importance of the obtained research results.What are the implications of these research findings for potential industrial application?Are the experimental methods used in these studies applied in industry?The research results should be summarized and some solutions or planned research should be indicated for the near future.
  • Pages 4 – 8, lines 102 – 164: Is it possible to estimate the costs of the applied research methods on an industrial scale?
  • Pages 4 – 8, lines 102 – 164: If it is possible, it is recommended to carry out a statistical analysis of the research results.
  • Page 9, line 172: The word ‘tht’ should be corrected.
  • Pages 1 – 9, lines 11 – 181: It is recommended to check the entire text for correctness of English language by an English philology specialist or a native speaker.
  • Page 9, lines 185 - 214 (References): The references should be improved. Journal abbreviations should be used instead of their full names. The number of references is a bit small and should be extended. Please, correct them and complete the information in accordance with the editorial requirements of the International Journal of Environmental Research and Public Health journal.

Round 2

Reviewer 3 Report

Thank you for improving the article according to reviewers' suggestions.